# Predictors of Loss to Follow-Up in an HIV Vaccine Preparedness Study in Masaka, Uganda

**DOI:** 10.3390/ijerph19116377

**Published:** 2022-05-24

**Authors:** Anita Kabarambi, Sheila Kansiime, Sylvia Kusemererwa, Jonathan Kitonsa, Pontiano Kaleebu, Eugene Ruzagira

**Affiliations:** 1Medical Research Council/Uganda Virus Research Institute and London School of Hygiene and Tropical Medicine Uganda Research Unit, Entebbe P.O. Box 49, Uganda; sheila.kansiime@mrcuganda.org (S.K.); sylvia.kusemererwa@mrcuganda.org (S.K.); jonathan.kitonsa@mrcuganda.org (J.K.); pontiano.kaleebu@mrcuganda.org (P.K.); eugene.ruzagira@mrcuganda.org (E.R.); 2London School of Hygiene and Tropical Medicine, London WC1E 7HT, UK

**Keywords:** HIV prevention, loss to follow up, HIV vaccine studies, high-risk, Uganda

## Abstract

Background: High participant retention is essential to achieve adequate statistical power for clinical trials. We assessed participant retention and predictors of loss to follow-up (LTFU) in an HIV vaccine-preparedness study in Masaka, Uganda. Methods: Between July 2018 and March 2021, HIV sero-negative adults (18–45 years) at high risk of HIV infection were identified through HIV counselling and testing (HCT) from sex-work hotspots along the trans-African highway and fishing communities along the shores of Lake Victoria. Study procedures included collection of baseline socio-demographic data, quarterly HCT, and 6-monthly collection of sexual risk behaviour data. Retention strategies included collection of detailed locator data, short clinic visits (1–2 h), flexible reimbursement for transport costs, immediate (≤7 days) follow-up of missed visits via phone and/or home visits, and community engagement meetings. LTFU was defined as missing ≥2 sequential study visits. Poisson regression models were used to identify baseline factors associated with LTFU. Results: 672 participants were included in this analysis. Of these, 336 (50%) were female and 390 (58%) were ≤24 years. The median follow-up time was 11 months (range: 0–31 months). A total 214 (32%) participants were LTFU over 607.8 person-years of observation (PYO), a rate of 35.2/100 PYO. LTFU was higher in younger participants (18–24 years versus 35–45 years, adjusted rate ratio (aRR) = 1.29, 95% confidence interval (CI) 0.80–2.11), although this difference was not significant. Female sex (aRR = 2.07, 95% CI, 1.51–2.84), and recreational drug use (aRR = 1.61, 95% CI, 1.12–2.34) were significantly associated with increased LTFU. Engagement in transactional sex was associated with increased LTFU (aRR = 1.36, 95% CI, 0.97–1.90) but this difference was not significant. LTFU was higher in 2020–2021 (the period of COVID-19 restrictions) compared to 2018–2019 (aRR = 1.54, 1.17–2.03). Being Muslim or other (aRR = 0.68, 95% CI 0.47–0.97) and self-identification as a sex worker (aRR = 0.47, 95% CI, 0.31–0.72) were associated with reduced LTFU. Conclusion: We observed a high LTFU rate in this cohort. LTFU was highest among women, younger persons, recreational drug users, and persons who engage in transactional sex. Efforts to design retention strategies should focus on these subpopulations.

## 1. Introduction

HIV remains a global public health problem with 1.5 million new infections reported in 2020 worldwide [1]. Most of the infections occur in sub-Saharan Africa, which accounts for 58% of all new HIV infections [1]. This is despite the increasing range of HIV prevention tools, which include male condoms, safe medical male circumcision, treatment as prevention, and oral pre-exposure prophylaxis (PrEP) [2].

A prophylactic HIV vaccine is still considered the best hope for controlling the epidemic. Several HIV vaccine-preparedness studies have been established in sub-Saharan Africa to describe the HIV epidemiology and assess the suitability and willingness of different populations to participate in future vaccine trials [3,4]. A critical goal of these studies is to demonstrate the ability to recruit and retain individuals at high risk of HIV infection in follow-up over long periods, as would be required in actual vaccine trials [5]. Loss to follow-up (LTFU) introduces bias in cohort studies [6]. This is because volunteers who get LTFU might have different characteristics from those who complete all study visits. This could result in either under- or over-estimation of the incidence of the outcome measure [5].

Understanding the predictors of LTFU is critical for informing strategies aimed at maximizing retention. Some HIV vaccine-preparedness studies in Uganda have assessed participant retention and factors associated with study dropout rate [5,7]. However, these studies have mainly been conducted among individuals from fishing communities [5,7,8]. We assessed predictors of loss to follow-up (LTFU) in an HIV vaccine-preparedness cohort study that enrolled individuals at high-risk of HIV infection from varied geographical and occupational backgrounds.

## 2. Methods

### 2.1. Study Design and Population

We used data from an ongoing HIV vaccine-preparedness cohort study, “The PrEPVacc Registration Cohort Study”. The study was set up at sites in Tanzania, Mozambique, South Africa, and Uganda to prepare a population of HIV-negative individuals at risk of acquiring HIV for possible participation in the PrEPVacc HIV vaccine efficacy trial [9,10].

In Uganda, the study is conducted at the Medical Research Council/Uganda Virus Research Institute and the London School of Hygiene and Tropical Medicine (MRC/UVRI and LSHTM) Uganda Research Unit’s clinical research site in Masaka city, Masaka district. Study participants were recruited from sex-work hotspots along the trans-African highway and fishing communities along the shores of Lake Victoria in Masaka and the neighboring districts of Rakai, Kalungu, Lwengo, and Lyantonde, an area spanning a radius of approximately 80 km from Masaka city [9]. To be eligible for the study, individuals had to be 18–45 years of age, HIV-negative, willing to provide locator information, available for follow-up, and have at least one of the following HIV risk indicators: suspected/confirmed sexually transmitted infection (STI), unprotected sex with ≥2 partners, unprotected sex with a new partner in the past 3 months, or unprotected transactional sex (giving/receiving money/goods in exchange for sex) in the past month.

#### Definitions

Missed visit: This was defined as a scheduled study follow-up visit whose ±7-day window closed without the participant attending the visit, and with no late visit attendance within a 3-month period from the scheduled visit.

Late visit: This was defined as a visit that occurred after the visit window for a scheduled follow-up visit had closed but before the window for the next scheduled follow-up visit opened.

LTFU: This was defined as missing two or more consecutive scheduled study visits and failure to attend further study visits after at least three documented attempts.

### 2.2. Procedures

#### 2.2.1. Identification and Recruitment of Participants

Prior to the initiation of the study, the study team held meetings with the community advisory board and community leaders from target communities to inform them about the study and obtain their input into the design of study tools and implementation of planned activities. Following these meetings, study team counsellors offered community-based HIV counselling and testing (HCT) services as per the national HIV testing algorithm [11].

Individuals who tested HIV-negative were provided with brief information about the study, and those willing were invited to the study clinic for screening and possible enrolment. At the screening visit, individuals were provided with more detailed study information and asked to provide written informed consent. Consenting individuals underwent repeat HCT, urine pregnancy testing (women), eligibility assessment, and enrolment if eligible. Enrolment procedures consisted of collection of participant locator information (participant’s names and addresses; names and contact details of next of kin or other persons who could be contacted for the purpose of tracing the participant; and permission for the research team to contact these persons). Information was also collected on socio-demographics as well as HIV risk and risk-reduction behaviours.

#### 2.2.2. Follow-Up of Participants

Participants were followed up every 3 months (one month = 28 days). Follow-up visit dates were estimated from the day of enrolment with a window of ±7 days for each visit. HCT was performed at each visit. Urine pregnancy testing (women) and collection of HIV risk and risk-reduction behaviour data were performed every 6 months. Retention strategies included: collection of detailed participant locator information at baseline, which was updated as necessary at follow-up visits; issuance of participant appointment cards with a detailed study visit schedule at the enrolment visit; use of phone calls, where possible, to remind participants of imminent or missed study visits; use of locator data to physically visit participants who were not contactable by phone and remind them to attend study visits; ensuring that study visits were as short (1–2 h) as practically possible to minimize study fatigue and allow participants to attend to other obligations; flexibility in reimbursing transport costs to accommodate changes in transport fares due to a change of residence or during known periods of peak demand on public transportation services; offer of psycho-social support and medical care for common illnesses to participants, their partners, and children; and community engagement meetings.

### 2.3. Statistical Methods

Data were entered and managed in OpenClinica. Analysis was conducted in STATA (College Station, TX, USA, version 15.0). Participant baseline characteristics were summarized using frequencies and percentages. The LTFU rate was estimated as the number of participants who were LTFU, divided by the total person-years of observation (PYO), expressed as a rate per 100 PYO. LTFU rates were reported overall, by demographic characteristics, and by calendar period i.e., 2018–2019 (pre-COVID-19) and 2020–2021 (COVID-19 restrictions). PYO were defined as time from enrolment until the last attended visit. For participants who were LTFU, the last attended visit was the visit prior to being categorised as LTFU. Participants who were not yet due for their first follow up visit were excluded from the analysis. We included follow-up data through March 2021.

Univariable and multivariable Poisson regression models were fitted to identify factors associated with LTFU. Factors associated with LTFU in HIV prevention studies were identified (from literature) and investigated within the data available. All variables assessed at univariable analysis were initially included in the multivariable analysis model. Variables were retained in the multivariable model if they had a *p*-value of ≤0.10 using backward elimination (Wald test). Sex and age were included and retained in the multivariable model a priori. Rate ratios with 95% confidence intervals (CI) were reported for both univariable and multivariable analyses. Kaplan–Meier plots were also used to assess associations with LTFU over time.

### 2.4. Ethical Considerations

The study protocol was approved by the Uganda Virus Research Institute Research Ethics Committee (reference number: GC/127/18/03/637) and the Uganda National Council for Science and Technology (reference number: HS2392). Prior to conducting study procedures, written informed consent was obtained from all participants. Individuals who tested HIV-positive during study follow-up were provided with post-test counselling and referred for HIV care. Pregnant HIV-positive female participants were also referred for services to prevent of mother-to-child HIV transmission.

## 3. Results

### 3.1. Baseline Socio-Demographic Characteristics

Of 1017 individuals who were screened for eligibility between July 2018 and March 2021, 728 (72%) were enrolled. The commonest reasons for ineligibility were being at low risk for HIV infection (n = 278, 96%), HIV infection (n = 5, 2%) and not being available for follow-up (n = 5, 2%). Of those enrolled, 672 (92%) had follow-up data and were included in the LTFU analysis. Of these, 336 (50%) were female, 390 (58%) were aged ≤24 years, 365 (54%) were single, and 403 (60%) had at most a primary-school-level education. (Table 1). Participants who were not yet due for their first follow-up visit (excluded from the analysis) were more likely to be female (84% vs. 50%, *p* < 0.001) and to self-identify as commercial sex workers (46% vs. 21%, *p* < 0.001) compared to those who were included in the analysis.

### 3.2. LTFU and Associated Factors

The median time of follow-up was 11 months (range: 0–31 months). A total 214 (31.8%) participants were LTFU over 607.8 PYO, an overall LTFU rate of 35.2/100 PYO. Of those LTFU, 78 (36%) did not attend any follow-up visit, or only attended one follow-up visit. The median time to LTFU for women was 1.5 years, while that for men was more than 2.5 years (Figure 1). The commonest reasons for LTFU were: being uncontactable (38%), moving out of the study area (27%), and withdrawing from the study (31%). Four participants died during follow up. LTFU was highest among participants that were aged 18–24 years (41.8/100 PYO, female (44.7/100 PYO), Christian (38.3/100 PYO), those working as salon/bar workers/street vendors (48.7/100 PYO), and in the 2020–2021 calendar period (42.6/100 PYO). There was weak evidence that LTFU was associated with age, with participants aged 18–24 years more likely to be LTFU (aRR = 1.29, 95% CI 0.80–2.11) compared with those aged 35–45 years. Female sex (adjusted rate ratio (aRR) = 2.07, 95% CI 1.51–2.84) and recreational drug use (aRR = 1.61, 95% CI 1.12–2.34) were significantly associated with increased LTFU. Transactional sex in the past month was associated with increased LTFU (aRR = 1.36, 95% CI 0.97–1.90) but this difference was not significant. LTFU was higher in 2020–2021 compared to 2018–2019 (aRR = 1.54, 1.17–2.03). LTFU was lower in Muslims (or other) than Christians (aRR = 0.68, 95% CI 0.47–0.97) and in participants who reported sex work as an occupation compared to those who did not (aRR = 0.47, 95% CI 0.31–0.72). Participants who reported abnormal genital discharge in the past 3 months had lower LTFU than those who did not (aRR = 0.79, 95% CI 0.59–1.04), but this difference was not significant (Table 1).

## 4. Discussion

We found a LTFU rate of 35.2 per 100 PYO in this HIV vaccine-preparedness cohort study. This level of LTFU is higher than levels reported in previous HIV vaccine-preparedness studies in Uganda in which annual LTFU ranged between 15% and 30% [5,7,8,12]. Some of these earlier studies were mostly conducted in relatively homogenous populations from small fairly well-defined geographical areas with the study procedures conducted within participants’ communities [8] or at study clinics that were located not too far away (10–40 km) [5,7]. In contrast, our study was conducted in a more heterogeneous population spread over a wide geographical area (up to 80 km from the study clinic) [10]. The national COVID-19 control measures that were instituted from March 2020 onwards also contributed to the high LTFU observed in this study. Indeed, LTFU was significantly higher in 2020–2021 compared to 2018–2019. The measures included stay-at-home orders and restrictions on public transportation for prolonged periods [13]. Although exceptions were made for clinical research activities to continue during these periods, participants still found it difficult to attend study visits as most depended on public transportation to move to and from the study site.

Consistent with previous studies [7], we found that a fairly large proportion (36%) of participants LTFU did not attend any follow-up visit, or only attended one follow-up visit. This finding lends support to proposals for preparatory cohorts in which potential HIV prevention trial participants demonstrate availability for long-term follow-up and willingness to comply with the study visit schedules prior to enrolment [7,14].

Although LTFU was higher among younger participants than their older counterparts, the association was not statistically significant. Previous HIV vaccine-preparedness studies found LTFU to be highest in the youngest age groups [5,7,8,15,16]. This is probably because older persons are likely to have stable employment, relationships, and residential status, factors that are associated with retention [17]. The finding that LTFU was significantly higher in women compared to men has been reported in some studies [7] but not others where LTFU was either higher in men compared to women [5] or did not differ by sex [8]. High LFTU among women may be attributed to competing responsibilities that commonly fall to women, such as caring for the sick, and childrearing [7]. Also, in our setting as in other areas in sub-Saharan Africa, retention of women may be affected by the fact that they have limited autonomy and are often disadvantaged economically [18,19,20,21,22]. For example a married woman often needs to obtain her partner’s permission to travel away from home as well as his financial support to cover travel expenses [18,19].

Drug abuse and commercial sex are associated with life instability that may be associated with poor retention in longitudinal studies [17]. This probably explains our finding that recreational drug use and reported transactional sex were associated with increased LTFU. Although 81% of our female study participants reported engaging in transactional sex, only 21% self-identified as sex workers. Individuals who self-identify as sex workers may be different from those who do not with respect to important factors that may affect LTFU. For example, in a study in Burkina Faso, self-identified sex workers tended to be professionals who worked from designated locations, while non-professional or indirect sex workers tended to be mobile street-side vendors, bar waitresses, students, and bar workers [23]. These differences may explain why participants who self-identified as sex workers in our study were less likely to be LTFU than those who did not. We found that LTFU was significantly lower among Muslim participants compared to Christian participants. The reasons for this are unclear. It is possible that religion is associated with other unmeasured factors that could impact retention.

A strength of this study is that participants were recruited from diverse backgrounds and geographical areas; hence the results may be more generalisable to a wider group of potential HIV prevention trial participants in this setting than those from more homogenous populations. One of the limitations of the study is that data were obtained by self-report. Hence, there is a possibility of response bias, particularly for stigmatising behaviours, e.g., transactional sex and use of recreational drugs. We did not collect data on distance to the study clinic, and therefore could not evaluate its impact on participant retention. However, other studies have identified distance to the study clinic as a significant barrier to retention [24,25,26]. We also did not interview participants who were LTFU, therefore, definite reasons for LTFU could not be established.

## 5. Conclusions

We observed a substantial LTFU rate in this HIV vaccine-preparedness study despite using several retention strategies. LTFU was more common among sub-populations that are likely to have the highest risk of HIV infection, i.e., women, younger persons, recreational drug users and those engaging in transactional sex. As high-risk individuals are the primary target of HIV prevention studies, retention strategies that target these groups should be urgently designed and evaluated.

## Figures and Tables

**Figure 1 ijerph-19-06377-f001:**
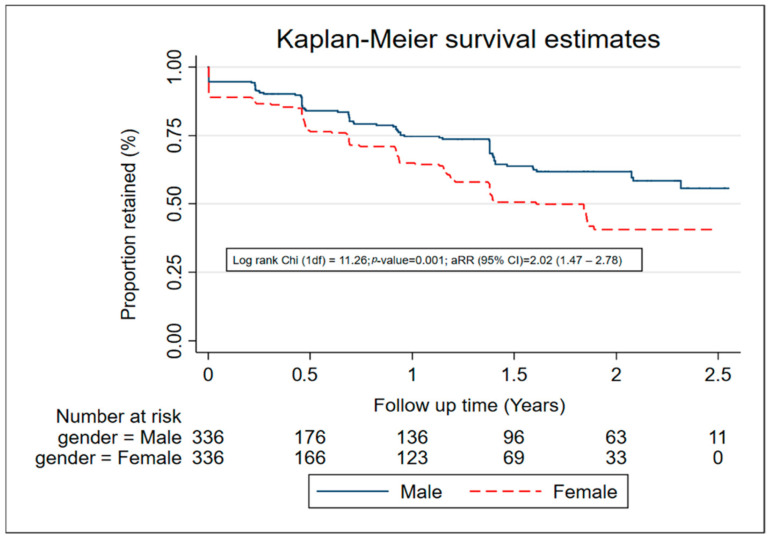
Retention among 672 participants in an HIV vaccine-preparedness cohort, Masaka, Uganda (July 2018–March 2021).

**Table 1 ijerph-19-06377-t001:** Loss to follow-up and associated factors among 672 participants in an HIV vaccine-preparedness cohort, Masaka, Uganda (July 2018–March 2021).

Characteristic			PYO	LTFU/100PYO (95% CI)	Bivariate Analysis		Multivariable Analysis	
	N (%)	n			uRR (95% CI)	*p*-Value	aRR (95% CI)	*p*-Value
Overall	672 (100)	214	607.8	35.2 (30.8–40.2)				
Gender								
Male	336 (50)	89	328.4	27.1 (22.0–33.4)	Ref		Ref	
Female	336 (50)	125	279.4	44.7 (37.5–53.3)	1.65 (1.26–2.17)	0.001	2.07 (1.51–2.84)	0.001
Age category								
35–45	49 (7)	19	64.8	29.3 (18.7–46.0)	Ref		Ref	
25–34	233 (35)	67	236.7	28.3 (22.2–36.0)	0.97 (0.58–1.61)		0.96 (0.57–1.60)	
18–24	390 (58)	128	306.3	41.8 (35.1–49.7)	1.42 (0.88–2.31)	0.024	1.29 (0.80–2.11)	0.122
Calendar period								
2018–2019 (pre-COVID-19)	405 (60) ^≠^	90	316.7	28.4 (23.1–34.9)	Ref		Ref	
2020–2021 (COVID-19 restrictions)	572 (85) ^≠^	124	291.1	42.6 (35.7–50.8)	1.50 (1.14–1.97)	0.003	1.54 (1.17–2.03)	0.002
Marital Status								
Single	365 (54)	104	292.5	35.6 (29.3–43.1)	Ref			
Married/cohabiting/in a relationship	213 (32)	76	244.4	31.1 (24.8–38.9)	0.87 (0.65–1.18)			
Divorced/widowed/separated	94 (14)	34	70.9	47.9 (34.2–67.1)	1.35 (0.92–1.99)	0.111	-	-
Education level								
≤Primary school	403 (60)	135	402.1	33.6 (28.4–39.7)	Ref			
≥Secondary school or higher	269 (40)	79	205.7	38.4 (30.8–47.9)	1.14 (0.87–1.51)	0.343	-	-
Religion								
Christian	509 (76)	177	462.2	38.3 (33.1–44.4)	Ref		Ref	
Muslim/ other	163 (24)	37	145.6	25.4 (18.4–35.1)	0.66 (0.47–0.95)	0.023	0.68 (0.47–0.97)	0.035
Occupation ^¶^								
Subsistence fisheries worker								
No	576 (86)	171	487.9	35.1 (30.2–40.4)	Ref			
Yes	96 (14)	43	119.9	35.9 (26.6–48.4)	1.02 (0.73–1.43)	0.893	-	-
Sex worker								
No	531 (79)	182	501.9	36.3 (31.3–41.9)	Ref		Ref	
Yes	141 (21)	32	105.9	30.2 (21.4–42.7)	0.83 (0.57–1.21)	0.340	0.47 (0.31–0.72)	0.001
Salon/bar worker/street vendor								
No	482 (72)	136	447.6	30.4 (25.7–35.9)	Ref			
Yes	190 (28)	78	160.2	48.7 (39.0–60.8)	1.60 (1.21–2.11)	0.001	-	-
Other occupation ^#^								
No	386 (57)	141	348.9	40.4 (34.3–47.7)	Ref			
Yes	286 (43)	73	258.9	28.2 (22.4–35.5)	0.70 (0.53–0.93)	0.013	-	-
Baseline HIV risk characteristics								
Used condom during last sex	172 (26)	62	145.0	42.8 (33.3–54.9)	1.30 (0.97–1.75)	0.080	-	-
Had transactional sex in the past month	547 (81)	166	428.8	38.7 (33.2–45.1)	1.44 (1.05–1.99)	0.025	1.36 (0.97–1.90)	0.073
Anonymous/casual sexual partners	638 (95)	195	564.2	34.6 (30.0–39.8)	0.79 (0.49–1.27)	0.332	-	-
Number of partners								
≤5	352 (52)	148	401.6	36.9 (31.4–43.2)	Ref			
≥6	320 (48)	66	206.2	32.0 (25.1–40.8)	0.87 (0.65–1.16)	0.342		
Abnormal genital discharge in the past 3 months	335 (50)	110	316.3	34.8 (28.8–41.9)	0.97 (0.74–1.27)	0.850	0.79 (0.59–1.04)	0.096
Genital sores/ulcers in the past 3 months	179 (27)	50	163.6	30.6 (23.2–40.3)	0.83 (0.60–1.14)	0.241	-	-
Used recreational drugs in the past 3 months	121 (18)	37	86.6	42.7 (30.9–58.9)	1.26 (0.88–1.79)	0.205	1.61 (1.12–2.34)	0.011

N = sample size; n = number with outcome (LTFU); PYO = person years of observation; LTFU = loss to follow-up; CI = confidence interval; uRR = unadjusted rate ratio; aRR = adjusted rate ratio; ^≠^ Total N may exceed 672 as some individuals contributed person-time to more than one calendar period; ^¶^ more than one option allowed; ^#^ other included: professional/technical worker, sales/service worker, motorcycle driver, house help, subsistence agricultural worker, office clerk, student, craft and related trades worker.

## Data Availability

Due to potentially identifying information in the data and the judgement of the LSHTM’s research quality management, all relevant data will be available upon request to interested, qualified researchers. Interested parties should submit a data request via the web form at https://doi.org/10.17037/DATA.00002915 (accessed on 28 March 2022) in the first instance. This request will be emailed to the LSHTM RDM Service and corresponding author for action. Requests can be sent directly to researchdatamanagement@lshtm.ac.uk (accessed on 28 March 2022), but this should be a secondary option, since we’ve found that requesters often don’t provide the correct information. Procedures for handling access requests can be found at https://researchonline.lshtm.ac.uk/id/eprint/612422/229/Research_Data_Management_Policy.pdf (accessed on 28 March 2022).

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
