# Peer review of "Predictors of Loss to Follow-Up in an HIV Vaccine Preparedness Study in Masaka, Uganda"

_ijerph, 2022, doi:10.3390/ijerph19116377_

Round 1

Reviewer 1 Report

This is a well written, concise and clear paper, documenting various contributing factors to loss to follow-up. The literature review is a bit light.

Perhaps a little more digging could be done to elucidate two factors pointed out by the authors.

Could the study team unearth the locations of people's residences, and hence distance to the clinic,  using tracking information, because it is an important potential factor and cannot be analyzed from questionnaire data, as the authors point out.  This factor would be relatively easy to remedy in a trial situation, by operating clinic hubs.

Given the known timing of the COVID-inspired lockdown, is it possible to evaluate before and after data to understand how important this factor was?

One factor is a little confusing-- self-identification as a sex worker is reported as a positive  factor in retention but transactional sex has the opposite effect, and sex might be a confounding factor here (e.g. is the person taking or giving payment?).  Perhaps this could be unpacked a bit.

very minor but in 2.2.2, I think followed up (verb) should not be hyphenated, though follow-up (noun) should be.

Author Response

Comments from Reviewer 1 and responses

Comment 1: This is a well written, concise and clear paper, documenting various contributing factors to loss to follow-up.

 Response: We would like to appreciate the reviewer for this compliment.

Comment 2: The literature review is a bit light. Perhaps a little more digging could be done to elucidate two factors pointed out by the authors. Could the study team unearth the locations of people's residences, and hence distance to the clinic, using tracking information, because it is an important potential factor and cannot be analysed from questionnaire data, as the authors point out.  This factor would be relatively easy to remedy in a trial situation, by operating clinic hubs.

Response: We thank the reviewer for this comment. We agree with the reviewer that using clinic hubs could potentially address the issue of distance to a research site for participants. We have indeed used such hubs for clinical studies where this is feasible. However, operating clinic hubs would be very challenging when evaluating HIV vaccine efficacy trials. These trials require sophisticated clinical and laboratory facilities for sample processing and storage and management of candidate vaccines. Thus it may not be feasible to establish these facilities in remote communities from where participants are recruited. It is partly for this reason that we are conducting HIV vaccine feasibility studies i.e., to evaluate the ability to recruit and retain participants from locations that are distant from the research sites.

We also agree that it would have been good to determine the association between distance to/from the research site and retention. We did not collect these data however. To address the transportation challenges, we reimbursed transport costs for each study visit. We also collected detailed locator information to ensure that participants could be tracked to their homes to remind them of study visits (if uncontactable by phone) or to find out reasons for withdrawal for those who did not intend to continue in follow-up.

Comment 3: Given the known timing of the COVID-inspired lockdown, is it possible to evaluate before and after data to understand how important this factor was?

Response: Thank you for this comment. We have conducted additional analyses to investigate the effect of COVID-19 restrictions of participant loss to follow-up (LTFU) by examining LTFU in for the period 2018-2019 (pre-COVID-19) and 2020-2021 (COVID-19 restrictions).  LTFU was significantly higher during the latter period. We have provided the results in the revised manuscript (abstract, Results section, Table 1) and updated the discussion

Comment 4: One factor is a little confusing-- self-identification as a sex worker is reported as a positive factor in retention but transactional sex has the opposite effect, and sex might be a confounding factor here (e.g. is the person taking or giving payment?).  Perhaps this could be unpacked a bit.

Response:   Thank you for this comment. Transactional sex was defined as giving or receiving gifts in exchange for sex. This has been clarified in revised manuscript (methods, section 2.1). We noted that not all people who engage in transactional sex self-identify as sex workers. In the discussion, we highlight that “Although 81% of our female study participants reported engaging in transactional sex, only 21% self-identified as sex workers.” 

We also noted that individuals who self-identify as sex workers may be different from those who do not with respect to important factors that may affect LTFU.  For example, in a study in Burkina Faso, self-identified sex workers tended to be professionals who worked from designated locations while non-professional or indirect sex workers tended to be mobile street-side vendors, bar waitresses, students, and bar workers. These differences may explain why participants who self-identified as sex workers in our study were less likely to be LTFU than those who did not.

We agree that sex could be a confounder and we did control for sex at multivariable analysis.

Comment 5: very minor but in 2.2.2, I think followed up (verb) should not be hyphenated, though follow-up (noun) should be.

Response: The hyphen on followed up has been removed.

Reviewer 2 Report

Predictors of loss to follow-up in an HIV vaccine preparedness study in Masaka, Uganda

Background: High participant retention is essential to achieve adequate statistical power for clinical trials. We assessed participant retention and predictors of loss to follow-up (LTFU) in an HIV vaccine preparedness study in Masaka, Uganda. Methods: Between July 2018 and March 2021, HIV sero-negative adults (18-45 years) at high risk of HIV infection were identified through HIV counselling and testing (HCT) from sex work hotspots along the trans-African highway and fishing communities along the shores of Lake Victoria. Study procedures included collection of baseline socio-demographic data, quarterly HCT, and 6-monthly collection of sexual risk behaviour data. Retention strategies included collection of detailed locator data, short clinic visits (1-2 hours), flexible reimbursement for transport costs, immediate (≤7 days) follow-up of missed visits via phone and/or home visits, and community engagement meetings. LTFU was defined as missing ≥2 sequential study visits. Poisson regression models were used to identify baseline factors associated with LTFU. Results: 672 participants were included in this analysis. Of these, 336 (50%) were female and 390 (58%) were ≤24 years. The median follow-up time was 11 months (range: 0-31 months). 214 (32%) participants were LTFU over 607.8 person-years of observation (PYO), a rate of 35.2/100 PYO. LTFU was higher in younger participants [18-24 years versus 35-45 years, adjusted rate ratio (aRR)=1.35, 95% confidence interval (CI) 0.83-2.19], although this difference was not significant. Female sex (aRR=2.02, 95% CI, 1.47–2.78), recreational drug use (aRR=1.59, 95% CI, 1.10–2.29) and engagement in transactional sex (aRR=1.42, 95% CI, 1.01–1.98) were significantly associated with increased LTFU. Self-identification as a sex worker was associated with reduced LTFU (aRR=0.49, 95% CI, 0.32-0.75). Conclusion: We observed a high LTFU rate in this cohort. LTFU was highest among women, younger persons, recreational drug users and persons who engage in transactional sex. Retention strategies that target these groups should be designed and evaluated.

Comments:

Of 1017 individuals who were screened for eligibility between July 2018 and March 2021, 728 (72%) were enrolled. The commonest reasons for ineligibility were being at low risk for HIV infection (n=278, 96%), HIV infection (n=5, 2%) and not being available for follow-up (n=5, 2%). Of those enrolled, 672 (92%) had follow-up data and were included in the LTFU analysis.

-is this a prospective or retrospective study? I cannot fully fathom the explanation and method

 Retention strategies that target these groups should be designed and evaluated.=  it is rather confusing because in this study Retention Strategy was tested. These included collection of detailed locator data, short clinic visits (1-2 hours), flexible reimbursement for transport costs, immediate (≤7 days) follow-up of missed visits via phone and/or home visits, and community engagement meetings.

  1. Methods

To be eligible for the study, individuals had to be 18-45 years of age, HIV-negative, willing to provide locator information and availa-ble for follow-up, and have at least one of the following HIV risk indicators: sus-pected/confirmed sexually transmitted infection (STI), unprotected sex with ≥2 partners, unprotected sex with a new partner in the past 3 months, or unprotected sex in exchange for money/goods in the past month.

Comment:How to confirm if the participant is really HIV-negative?

2.1. Study design and population

2.2. Procedures

2.2.1. Identification and recruitment of participants

2.2.2. Follow-up of participants

2.3. Definitions

2.4. Statistical methods

Comments: The Definitions should be under 2.1

  1. Results

3.1. Baseline socio-demographic characteristics

3.2. LTFU and associated factors

Comments: What about the results of  Retention Strategy?

Table 1. Loss to follow-up and associated factors among 672 participants in an HIV vaccine prepar-edness cohort, Masaka, Uganda (July 2018-March 2021).

Comment, N=672, what about n=214 ?

Of 1017 individuals who were screened for eligibility between July 2018 and March 2021, 728 (72%) were enrolled. The commonest reasons for ineligibility were being at low risk for HIV infection (n=278, 96%), HIV infection (n=5, 2%) and not being available for follow-up (n=5, 2%). Of those enrolled, 672 (92%) had follow-up data and were included in the LTFU analysis.

Figure 1. Retention among 672 participants in an HIV vaccine preparedness cohort, Masaka, Uganda (July 2018-March 2021).

The figure is incorrect. For instance, after 2.5 years zero female left, but this was not shown in the   Kaplan-Meier curve

strength of this study is that participants were recruited from diverse backgrounds and geographical areas; hence the results may be more generalizable than those from more homogenous populations.

Comment: this notion is not applicable unless you have done the power calculation/sample size estimation

  1. Conclusion

We observed a substantial LTFU rate in this HIV vaccine preparedness study despite using several retention strategies. LTFU was more common among sub-populations that are likely to have the highest risk of HIV infection i.e., women, younger persons, recrea-tional drug users and those engaging in transactional sex. Retention strategies that target these groups should be designed and evaluated.

Comment: pls strengthen the conclusion with more findings

Author Response

Comments from Reviewer 2 and responses

Comment 1: Of 1017 individuals who were screened for eligibility between July 2018 and March 2021, 728 (72%) were enrolled. The commonest reasons for ineligibility were being at low risk for HIV infection (n=278, 96%), HIV infection (n=5, 2%) and not being available for follow-up (n=5, 2%). Of those enrolled, 672 (92%) had follow-up data and were included in the LTFU analysis.

-is this a prospective or retrospective study? I cannot fully fathom the explanation and method

Response: Thank you for your comment. This is a prospective study. In section “2.1. Study design and population” we state that “We used data from an ongoing HIV vaccine preparedness cohort study…..” 

Comment 2: Retention strategies that target these groups should be designed and evaluated. =  it is rather confusing because in this study Retention Strategy was tested. These included collection of detailed locator data, short clinic visits (1-2 hours), flexible reimbursement for transport costs, immediate (≤7 days) follow-up of missed visits via phone and/or home visits, and community engagement meetings.

Response: We wish to clarify that study used several strategies to maximise retention but did not set out to formally evaluate these strategies. We evaluated retention in this context and found that despite these strategies, LTFU is high in certain sub-populations. These groups may require different or additional retention strategies that specifically target them. 

Methods

Comment 3: To be eligible for the study, individuals had to be 18-45 years of age, HIV-negative, willing to provide locator information and available for follow-up, and have at least one of the following HIV risk indicators: suspected/confirmed sexually transmitted infection (STI), unprotected sex with ≥2 partners, unprotected sex with a new partner in the past 3 months, or unprotected sex in exchange for money/goods in the past month.

How to confirm if the participant is really HIV-negative?

Response: Thank you for your comment. Study team counsellors offered community-based HIV counselling and testing (HCT) services as per the national HIV testing algorithm. Individuals who tested HIV-negative were provided brief information about the study and those willing, invited to the study clinic for screening and possible enrolment. “At the screening visit, individuals were provided more detailed study information and asked to provide written informed consent. Consenting individuals underwent repeat HIV counselling and testing, urine pregnancy testing (women), eligibility assessment, and enrolment if eligible. This information is provided in the section with study procedures. (Section 2.2) 

Comment 4: The Definitions should be under 2.1

Response: We thank the reviewer for pointing this out. We have moved the definitions from section 2.3 to section 2.1.

Results

Comment 5:

3.1. Baseline socio-demographic characteristics

3.2. LTFU and associated factors

Comments: What about the results of Retention Strategy?

Response: Please see response to comment 2 above

Comment 6: Table 1. Loss to follow-up and associated factors among 672 participants in an HIV vaccine preparedness cohort, Masaka, Uganda (July 2018-March 2021). N=672, what about n=214?

Response: We thank the reviewer for this question. n=214 is the number of participants who got lost to follow up. In Table 1, we have clarified that n=number is the of participants with outcome (LTFU). These participants are part of the 672.

Comment 7: Of 1017 individuals who were screened for eligibility between July 2018 and March 2021, 728 (72%) were enrolled. The commonest reasons for ineligibility were being at low risk for HIV infection (n=278, 96%), HIV infection (n=5, 2%) and not being available for follow-up (n=5, 2%). Of those enrolled, 672 (92%) had follow-up data and were included in the LTFU analysis. Figure 1. Retention among 672 participants in an HIV vaccine preparedness cohort, Masaka, Uganda (July 2018-March 2021).

The figure is incorrect. For instance, after 2.5 years zero female left, but this was not shown in the Kaplan-Meier curve

Response: We thank the reviewer for this observation. However, we wish to clarify that participants had varying time in follow up with some “censored” earlier than others. The overall median time of follow-up was 11 months (range: 0-31 months); for females it was 10 months (range 0-29.7 months). There not being females “at risk” at the 2.5-year mark is not a representation that they were all LTFU. Rather, there were no female participants making it to that time point (2.5-year mark) at the time of analysis. The graph correctly represents this.   We have added a sentence in the results section to show that the median time to LTFU for women was 1.5 years while that for men was more than 2.5 years (Figure 1).

Comment 8: Strength of this study is that participants were recruited from diverse backgrounds and geographical areas; hence the results may be more generalizable than those from more homogenous population. This notion is not applicable unless you have done the power calculation/sample size estimation.

Response: We appreciate the reviewer for this observation. However, we consider that having recruited individuals from diverse backgrounds/geographies, our results are more generalisable more generalizable to a wider group of potential HIV prevention trial participants in this setting than those from more homogenous populations e.g., fishing communities or female sex workers. We have clarified this in the revised draft.

Comment 9: Conclusion; We observed a substantial LTFU rate in this HIV vaccine preparedness study despite using several retention strategies. LTFU was more common among sub-populations that are likely to have the highest risk of HIV infection i.e., women, younger persons, recreational drug users and those engaging in transactional sex. Retention strategies that target these groups should be designed and evaluate. Please strengthen the conclusion with more findings

Response: We thank the reviewer for this comment. We consider that it is critical to highlight that the LTFU was highest among individuals with at the highest risk of HIV infection. These individuals are the primary target of HIV prevention studies. Hence it is critical that retention strategies that target these groups are urgently designed and evaluated.  We revised the conclusion to make this clearer.

Reviewer 3 Report

The subject of the manuscript is of paramount importance. However, the introduction does not justify the gap that highlights the importance of the study in light of other research. Therefore, the authors could make an effort not to leave a brief introduction on the subject and further develop the importance associated with the study.

On the other hand, I consider two important points to clarify about the statistical analysis process. a) Justify why they use a p-value of ≤0.10 using backward elimination (wald test)" why not a p-value of ≤0.05? Explain from the perspective of statistical rigor with respect to the critical value of alpha. b) in their Kaplan-Meier result, the results do not indicate the Chi-square, df, and p-value values, it is important to locate them in the results. A final point, the authors must indicate that they verified the associated assumptions of the statistical tests that they applied.

In this sense, one of the basic interpretations of the Kaplan-Meier test is to explain the comparison of the survival curves when reaching a proportion of 0.5. The authors do not show this argumentative contrast as a function of time in years. This should be exposed in the discussion since it is a concrete way of contrasting with other studies at that proportional level. In addition, table 1 can be improved, since it is very saturated. For example, the (95% CI) of the estimates indicate them as the two values, you could indicate the higher CI value and explain it in the title of the Table, this would help reduce numerical data content.

Author Response

Comments from Reviewer 3 and responses

Comment 1: The subject of the manuscript is of paramount importance. However, the introduction does not justify the gap that highlights the importance of the study in light of other research. Therefore, the authors could make an effort not to leave a brief introduction on the subject and further develop the importance associated with the study.

Response: We thank the reviewer for this comment. We have added the following information to the discussion to emphasize the importance of the subject. “Loss to follow-up (LTFU) introduces bias in cohort studies[6]. This is because volunteers who get LTFU might have different characteristics from those who complete all study visits. This could result in either under- or over-estimation of the incidence of the outcome measure. Understanding the predictors of LTFU is critical for informing strategies aimed at maximizing retention.”

Comment 2: On the other hand, I consider two important points to clarify about the statistical analysis process. a) Justify why they use a p-value of ≤0.10 using backward elimination (wald test)" why not a p-value of ≤0.05? Explain from the perspective of statistical rigor with respect to the critical value of alpha.

Response: Thank you for your comment. The choice of a p-value cut-off of ≤0.10 for variable selection in the model developed was largely arbitrary. We wish to further clarify that p-value cut off of 0.05 and 0.10 are considered to both be traditional options for variable selection cut-offs for model building. (Reference: Chowdhury MZI, Turin TC. Variable selection strategies and its importance in clinical prediction modelling. Fam Med Com Health 2020;8:e000262. doi:10.1136/ fmch-2019-000262

We chose the cut-off of p<0.1 (rather than 0.05) because the 5%-level is quite stringent (i.e. can be difficult to achieve especially with smaller datasets), and because there are no serious consequences of retaining an unimportant explanatory variable in the model since emphasis is to be placed on the effect estimate (the aRR in this case) not the P-Value.  More justification is provided for in the reference shared above.

Comment 3: In their Kaplan-Meier result, the results do not indicate the Chi-square, df, and p-value values, it is important to locate them in the results.

Response: We appreciate the review for this suggestion. These details have been added and Kaplan Meier plot updated as shown below.

Comment 4: A final point, the authors must indicate that they verified the associated assumptions of the statistical tests that they applied.

In this sense, one of the basic interpretations of the Kaplan-Meier test is to explain the comparison of the survival curves when reaching a proportion of 0.5. The authors do not show this argumentative contrast as a function of time in years. This should be exposed in the discussion since it is a concrete way of contrasting with other studies at that proportional level. In addition, table 1 can be improved, since it is very saturated. For example, the (95% CI) of the estimates indicate them as the two values, you could indicate the higher CI value and explain it in the title of the Table, this would help reduce numerical data content.

Response: Thank you for this question. We have now highlighted in the results that women were significantly more likely to get LTFU than men by including the following statement – “The median time to LTFU for women was 1.5 years while that for men was more than 2.5 years”

Regarding table 1, we opted to instead use the horizontal layout which is more spacious.  We are hesitant to only include the higher CI (as suggested) since the precision of the CI is useful to know.   

Round 2

Reviewer 2 Report

All comment have been well addressed